# Incidence and Mortality Trends of Upper Respiratory Infections in China and Other Asian Countries from 1990 to 2019

**DOI:** 10.3390/v14112550

**Published:** 2022-11-18

**Authors:** Qiao Liu, Chenyuan Qin, Min Du, Yaping Wang, Wenxin Yan, Min Liu, Jue Liu

**Affiliations:** 1Department of Epidemiology and Biostatistics, School of Public Health, Peking University, Beijing 100191, China; 2Institute for Global Health and Development, Peking University, Beijing 100871, China; 3Global Center for Infectious Disease and Policy Research, Global Health and Infectious Diseases Group, Peking University, Beijing 100191, China

**Keywords:** upper respiratory infections, China, Asia, burden, trends

## Abstract

Respiratory infections remain a major public health problem, affecting people of all age groups, but there is still a lack of studies analyzing the burden of upper respiratory infections (URIs) in Asian countries. We used the data from the Global Burden of Diseases Study 2019 results to assess the current status and trends of URI burden from 1990 to 2019 in Asian countries. We found that Thailand had the highest age-standardized incidence rate (ASIR) of URI both in 1990 (354,857.14 per 100,000) and in 2019 (344,287.93 per 100,000); and the highest age-standardized mortality rate (ASMR) was in China in 1990 (2.377 per 100,000), and in Uzbekistan in 2019 (0.418 per 100,000). From 1990 to 2019, ASIRs of URI slightly increased in several countries, with the speediest in Pakistan (estimated annual percentage change [EAPC] = 0.404%, 95% CI, 0.322% to 0.486%); and Kuwait and Singapore had uptrends of ASMRs, at a speed of an average 3.332% (95% CI, 2.605% to 4.065%) and 3.160% (95% CI, 1.971% to 4.362%) per year, respectively. The age structure of URI was similar at national, Asian and Global levels. Children under the age of five had the highest incidence rate, and the elderly had the highest mortality rate of URI. Asian countries with a Socio-demographic Index between 0.5 and 0.7 had relatively lower ASIRs but higher ASMRs of URIs. The declined rate of URI ASMR in Asian countries was more pronounced in higher baseline (ASMR in 1990) countries. Our findings suggest that there was a huge burden of URI cases in Asia that affected vulnerable and impoverished people’s livelihoods. Continuous and high-quality surveillance data across Asian countries are needed to improve the estimation of the disease burden attributable to URIs, and the best public health interventions are needed to curb this burden.

## 1. Introduction

Respiratory infections remain a major public health around the world. Infections of the upper respiratory tract, such as laryngitis, pharyngitis, nasopharyngitis, and rhinitis, are among the most common diseases in primary medical care [1,2]. Respiratory infections caused by influenza affect people in all age groups, resulting in 3 to 5 million cases of severe illness, and about 290,000 to 650,000 deaths annually across the world [3]. Lower respiratory infections (LRI) caused by environmental tobacco smoke exposure were also an important threat to people’s health, especially for children and infants [4,5,6]. As home to nearly 60% of the global population, and including three of the world’s most populous countries (China, India, and Indonesia) [7], Asia could be well positioned to exert a significant influence on the global prevention and control of respiratory infections.

The Global Burden of Disease Study 2019 (GBD 2019) results consisted of a systematic and scientific effort to quantify the comparative magnitude of health losses due to diseases by sex, age, and location over time [8]. There were comprehensive efforts that described global, regional, and national burden and trends of LRIs using data from GBD results [9,10,11]. Nevertheless, the high burden of upper respiratory infections (URIs) could not be neglected. URI accounted for billions of dollars in annual health care cost, and it might increase burden on health resources, as acute respiratory infections were the most common reason for acute care appointment [12]. Additionally, a recent study showed that previous URI histories were associated with increased COVID-19 susceptibility and morbidity [13]. A recent study had assessed the global burden of URIs, mostly discussing regional URIs, but had not emphasized the great URI burden in Asian countries [14]. These facts attached importance on the efforts to analyze the burden of URIs in Asian countries, which, together with previous studies focusing on the burden of LRIs, may help the world understand more about respiratory infections in Asia.

In this study, we aimed to assess the current status and trends of URI burden from 1990 to 2019 in Asian countries, using data from the GBD 2019 results. Additionally, despite the susceptibility in all age groups with URIs, there are groups that are more at risk than others [3]; thus, we chose China, which contains the largest population not only among Asian countries but also all over the world, to explore the age and sex disparities of URIs. Our study can provide a comprehensive perspective for better understanding of the long-term trends and national differences in incidence and mortality of URIs among Asian countries, which could serve as an extension and complement to previous studies to better understand respiratory infections, thus formulating making national and regional strategies for prevention and control of influenza viral infections and other factors that lead to respiratory infections.

## 2. Materials and Methods

### 2.1. Data Source

We used the data obtained from the GBD 2019 result tools, established by the GBD group [15]. We extracted annual data of incidence number, incidence rates, death number, and mortality rates of upper respiratory infections. The general methodological approaches to estimate the mortality were described elsewhere [16]. Vital registration and surveillance data from the cause of death database were used in a generic cause of death ensemble modelling approach to estimate the URI mortality; and data from national surveys were used for the estimation of URI cases, which were modeled by a standard disease model-Bayesian meta-regression [16].

We reported the incidence and death results of URI in China and other Asian countries, as well as the overall Asian and Global status of URI, from 1990 to 2019. Additionally, we arranged the incidence and death data into successive 5-year age intervals from <5 years to 75–79 years, plus the 80+ years group. Then the 5-year age groups were further grouped into five groups, including <5 years, 5 to 14 years, 15 to 49 years, 50 to 69 years, and 70 plus years in this study.

We also extracted the Socio-demographic Index (SDI) of Asian countries. The SDI is a composite indicator of development status strongly correlated with health outcomes. It is the geometric mean of 0 to 1 indices of total fertility rate under the age of 25, mean education for those ages 15 and older, and lag distributed income per capita. As a composite, a country with an SDI of 0 would have a theoretical minimum level of development relevant to health, while a country with an SDI of 1 would have a theoretical maximum level [17].

### 2.2. Statistical Analysis

Absolute incidence and death number represented the actual situation of URI in each country, and its relative change was defined as (Numbers_2019_ − Numbers_1990_)/Numbers_1990_×100%, which showed the overall change between 1990 and 2019. Age-standardized incidence rate (ASIR) and age-standardized mortality rate (ASMR), which were directly extracted from the GBD result tool, [15] were calculated by applying the age-specific rates to a GBD World Standard Population, and were used to compare populations with different age structures or for the same population over time in which the age profiles change accordingly.

Estimated annual percentage change (EAPC) was widely used to quantify the rate trend over a specific interval [18,19,20]. A regression line was fitted to the natural logarithm of the rates (*y* = *α* + *βx* + *ε*, where y = ln (rate) and x = calendar year). EAPC was calculated as 100 × (*e^β^* − 1), with 95% confidence intervals (CIs) obtained from the linear regression model. In this study, overall EAPC was calculated by the annual ASIR and ASMR of URI in Asian countries, and EAPC in different age groups was calculated by the age-specific incidence and mortality rate. The term “increase” was used to describe trends when the EAPC and its lower boundary of 95% CI were both >0. In contrast, “decrease” was used when the EAPC and its upper boundary of 95% CI, were both <0. Otherwise, the term “stable” was used.

Lastly, to explore the influential factors for ASRs and EAPCs, we conducted nonlinear regression (polynomial) to assess the association between EAPCs and ASRs (in 1990), and SDIs (in 2019), as well as the association between ASRs (in 2019) and SDIs (in 2019) among Asian countries.

## 3. Results

### 3.1. Global, Asian and National Burden and Trends of URI from 1990 to 2019

The ASIR of URI in China (189,716.13 in 1990 and 191,087.67 in 2019, per 100,000) was lower than both global and Asian level, yet the ASMR in China (2.377 per 100,000 in 1990, and 0.208 per 100,000 in 2019) was much higher than the global and Asian level. From 1990 to 2019, ASIRs were stable in China and Asia, while the global ASIR of URI slightly decreased at a speed of an average 0.077% (95% CI, 0.062% to 0.093%) per year. However, ASMRs decreased in China, Asia and worldwide, with the speediest decrease in China (EAPC = −9.671%, 95% CI, −10.295% to −9.043%). China respectively accounted for 33% and 68% of the Asian incidence and death number of URI in 1990, and 27% and 60% in 2019. The proportion that China accounted for was even appreciable at the global level: 18% and 44% of the Global incidence and death number of URI in 1990, and 15% and 30% in 2019 (Figure 1).

Among all Asian countries, Thailand had the highest ASIR of URI both in 1990 (354,857.14 per 100,000) and in 2019 (344,287.93 per 100,000), followed by Japan (ASIR = 302,648.50 in 1990 and 302,776.04 in 2019, per 100,000) and Turkey (ASIR = 294,809.62 in 1990 and 293,415.45 in 2019, per 100,000); and the highest ASMR was in China in 1990 (2.377 per 100,000), and in Uzbekistan in 2019 (0.418 per 100,000). From 1990 to 2019, ASIRs of URI slightly increased in several countries, with the speediest in Pakistan (EAPC = 0.404%, 95% CI, 0.322% to 0.486%), followed by Uzbekistan (EAPC = 0.024%, 95% CI, 0.004% to 0.045%) and Maldives (EAPC = 0.020%, 95% CI, 0.005% to 0.035%). While ASMR of URI in most Asian countries substantially decreased from 1990 to 2019, Kuwait and Singapore had uptrends of ASMRs, at a speed of an average 3.332% (95% CI, 2.605% to 4.065%) and 3.160% (95% CI, 1.971% to 4.362%) per year, respectively. The largest incidence number of URI was from China in 1990 (2.29 billion) and from India in 2019 (2.70 billion), while China had the largest death number of URI both in 1990 (15.1 thousand) and 2019 (2.8 thousand). Besides India, there are nine Asian countries with URI cases over 200 million in 2019 (China, Indonesia, Pakistan, Bangladesh, Japan, Philippines, Turkey, Thailand and Iran) (Figure 1).

### 3.2. Age and Sex Distribution of URI in China

The age structure of URI in China was similar to that at the Asian and Global levels. Children under the age of five had the highest incidence rate and the second highest mortality rate, while the highest mortality rate of URI was in people aged over 70. All the five age groups had stable incidence rates throughout 1990 to 2019, while the mortality rates decreased in the five age groups. Notably, although the incidence rates in China were lower than the Asian and global levels, the mortality rates were higher, in all the five age groups. (Figure 2). In China, male cases were more frequent than female cases from 1990 to 2019, however, there were more deaths in female patients in the 1990s than in male patients in the 2010s. The sex distribution at the Asian and global levels was the same as that in China. (Figure 3). From 1990 to 2019, the global incidence rates of URI decreased in all age groups, with the speediest in children under the age of five (EAPC = −0.15%, 95% CI, −0.21% to −0.10%); trends of URI incidence rates in Asia varied in different age groups, children aged 5–9 years had the incidence rates increased at a speed of an average 0.09% (95% CI, 0.02% to 0.16%) per year, while people aged 50–54 years had incidence rates decrease at a speed of an average 0.09% (95% CI, 0.07% to 0.11%) per year; the incidence rates of URI was stable in most age groups in China, except people aged 55–59 years (EAPC = −0.02%, 95% CI, –0.04% to −0.01%), 30–34 years (EAPC = −0.02%, 95% CI, −0.03% to −0.003%), and 25–29 years (EAPC = −0.01%, 95% CI, −0.02% to −0.001%). Meanwhile, all the age groups had the mortality rates decreased from 1990 to 2019 in China, and at Asian and global level, with the speediest in children under the age of 5 in China (EAPC = −11.64%, 95% CI, −12.19% to –11.08%) (Figure 4). From 1990 to 2019, among all URI cases and deaths, the proportion of people from 50–69 years and 70+ groups increased, while people under 15 years had decreased proportions during the past three decades. (Figure 5).

The age disparity of incidence and death number of URI was significant in China. In 1990, children under the age of five had the largest incidence number of URI (162 million men and 132 million women), and most URI cases were concentrated in people younger than 30. The largest death number of URI was also from children under the age of five (2837 men and 2947 women), followed by people older than 80 (1328 men and 2457 women), which had the smallest incidence number in 1990 (3.3 million men and 5.5 million women). In 2019, people in the 30–34 year-old group had the largest incidence number of URI (126 million men and 117 million women), and the smallest death number was from people older than 80 (13.7 million men and 20.2 million women), which, however, had the largest death number in 2019 (650 men and 904 women) (Figure 6).

### 3.3. Factors Associated with URI Burden and EAPCs

The ASIR of URI had a U-shaped association with SDI (in 2019) in Asian countries, with the peak point appearing SDI between 0.5 an 0.6; and the ASMR of URI had an inverted U-shaped association with SDI (in 2019) in Asian countries, with the peak point also appearing as SDI between 0.5 and 0.6. The association of EAPCs of ASIR with SDI (in 2019) or ASIR (in 1990) was not obvious. However, the EAPCs of ASMR had a U-shaped association with SDI (in 2019), with the peak point appearing as SDI between 0.6 and 0.7; and the EAPCs of ASMR had an overall downtrend when ASMR (in 1990) increased. (Figure 7).

## 4. Discussion

To the best of our knowledge, this is the first comprehensive effort to describe the current status of URIs in Asian countries, also assessing the long-term trends of URI burden throughout the past three decades and health disparities in age and sex; and exploring the association of the incidence and mortality rates of URIs with the SDI, as well as the association of the EAPCs with the SDI and the disease burden. India, China, Indonesia and seven other countries had more than 200 million URI cases in 2019. The highest ASIR of URIs was observed in Thailand, followed by Japan and Turkey. These huge numbers of URI cases indicated an extremely high burden of health resources, despite the relatively low death rates of URIs. From 1990 to 2019, ASIRs of URIs in Asian countries were stable or had slight increases (Pakistan, Uzbekistan and Maldives), while ASMRs of URIs in most Asian countries substantially decreased during the last three decades, except Kuwait and Singapore. The age and sex distribution of URI cases and deaths was consistent at the national, regional and global levels. Children under the age of five were more susceptible than other age groups, and the elderly had lower incidence rates but extremely higher mortality rates than other age groups. Sex disparity was small, with male cases and deaths slightly higher than the females in recent years, however, there were more female cases and deaths of URIs in people over 80 years. Asian countries with SDI between 0.5 and 0.7 had relatively lower ASIRs but higher ASMRs of URIs. The declined rate of URI ASMR in Asian countries was more pronounced in higher baseline (ASMR in 1990) countries. Facing the large burden and complex situation of URIs in Asia, multifaceted and multisectoral actions are needed in all Asian countries to formulate national vaccination policies and other strategies to reduce the burden on the medical system due to URIs, especially at the time when COVID-19 was still transmitting all over the world.

According to our results, there were Asian countries with significant number of URI cases such as in India, China and Indonesia, and countries with high ASIR of URIs such as Thailand, Japan and Turkey. It was reported that the rate of influenza vaccine sales in Southeast Asia remained low and failed to consist with global guidelines, which might lead to the high burden of URIs caused by influenza in Southeast Asian countries [21]. There also remained the lack of surveillance data on URIs from many areas of large landmass countries such as India [22]. Facing these challenges, international cooperation on infectious disease control among Asian countries was established, such as the cross-country management of infectious diseases in the Association of Southeast Asian Nations [23], the Global Influenza Initiative (a global expert scientific forum comprising scientists, researchers and clinicians with expertise in epidemiology, immunology, infectious diseases, and public health) [24], and Asia-Pacific Alliance for the Control of Influenza. [25]. For Asian countries to effectively control and protect URIs, annual vaccination is required, and enhanced and continued surveillance remains critical.

Our study showed great age disparities of URI cases and deaths, mostly affecting the livelihoods of children under the age of five and the elderly. It was reported that children under five years and the elderly could be at great risk of severe disease or complications when infected by influenza viruses [3]. A recent study also found that the detection of *H. influenzae* or *Klebsiella* spp. in the upper respiratory tract was associated with lower respiratory infections in children [26]. Most deaths associated with influenza were found to occur among individuals aged 65 or older in industrialized countries [27]. Furthermore, though sex disparity of URIs was relatively small in our study, there were more female cases and deaths of URIs in people over 80 years. This fact emphasized the special position of women in URIs, not only in pregnancy, but also as the elderly, since pregnant women, in general, are vulnerable to respiratory infections [28].

Asian countries with SDI between 0.5 and 0.7 had relatively lower ASIRs but higher ASMRs of URIs, indicating the modifying role of socioeconomic status in the effect of air pollution on respiratory tract infections. The ASMR of URIs had an inverted U-shaped association with SDI (in 2019) in our study, which might be due to the U-shaped association between ambient PM_2.5_ and socioeconomic development, with ambient PM_2.5_ exposure highest in countries with middle SDI [29]. A previous study conducted in Southeast China also showed that air particulate exposure could increase the risk of respiratory *Heamophilus influenzae* infection [30]. Additionally, an association of PM_2.5_ exposure with increased acute respiratory infection risk was also found in young children with proven respiratory syncytial virus infection [31]. With the adverse health effects of air pollution during the life course and on children’s livelihoods, policymakers need to be goaded into more action for the immediate rapid reduction in fossil fuel emissions to protect population health [32].

Besides air pollution, smoking and vaping also have an important association with respiratory tract infections. A previous study showed that most countries, which had significantly higher smoking prevalence than the global average, were located in southeast Asia; and China, India, and Indonesia had the three largest smoking populations [33]. Environmental tobacco smoke exposure could cause LRIs and affects people’s livelihoods, especially that of children and infants [4,5,6]. Studies showed that E-cigarette exposure could disrupt pulmonary homeostasis, by disturbing gas exchange [34], reducing lung function [35,36], increasing airway inflammation and oxidative stress, [37,38] downregulating immunity [39], and increasing risk of respiratory infection [40]. In our study, we also found that the declined rate of URI ASMR was more pronounced in higher ASMR baseline countries, probably because countries with a higher baseline ASMR, such as China, found it easier to reduce the massive URI outbreaks by establishing national public-health strategies, such as the vaccination campaign [41].

Our study had several limitations. First, data from GBD results were estimated by several models, and the real burden of URIs could be misestimated [16]. Nevertheless, our findings could alert the world that it is of vital importance to take national actions to control URIs, especially during events such as the COVID-19 pandemic. Second, due to the statistical characteristics of GBD results, data on certain pathogens that cause URIs were not reported.

## 5. Conclusions

There was a huge burden of URI cases in Asia that affected vulnerable and impoverished people’s livelihoods, especially for children under five years and the elderly, and also increased burden on health resources. Continuous and high-quality surveillance data across China and other Asian countries are needed to improve the estimation of the disease burden attributable to URIs and the best public health interventions, including vaccination policies are needed to curb this burden.

## Figures and Tables

**Figure 1 viruses-14-02550-f001:**
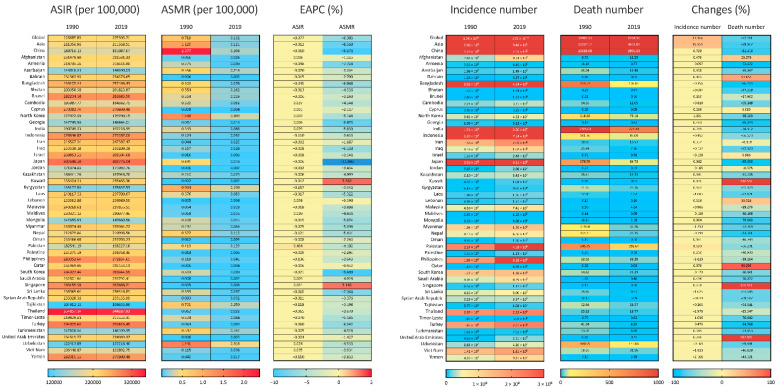
A heatmap of ASIR, ASMR, EAPC, and number and changes of upper respiratory infection cases and deaths between 1990 and 2019, in globe, Asia, and Asian countries. ASIR, age-standardized incidence rate; ASMR, age-standardized mortality rate; EAPC, estimated annual percentage change.

**Figure 2 viruses-14-02550-f002:**
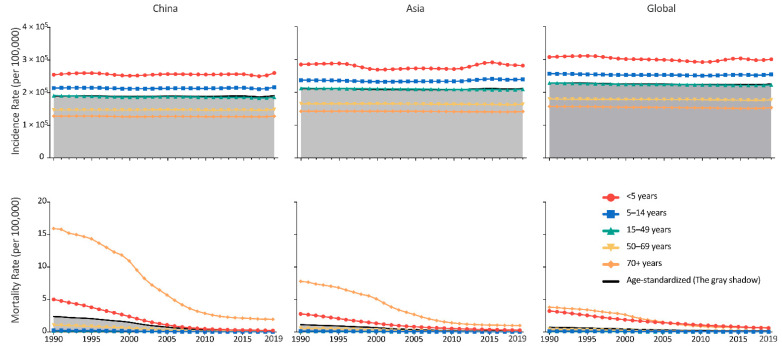
The incidence and mortality rate of upper respiratory infections by age group, from 1990 to 2019, in China, Asia, and the rest of the globe.

**Figure 3 viruses-14-02550-f003:**
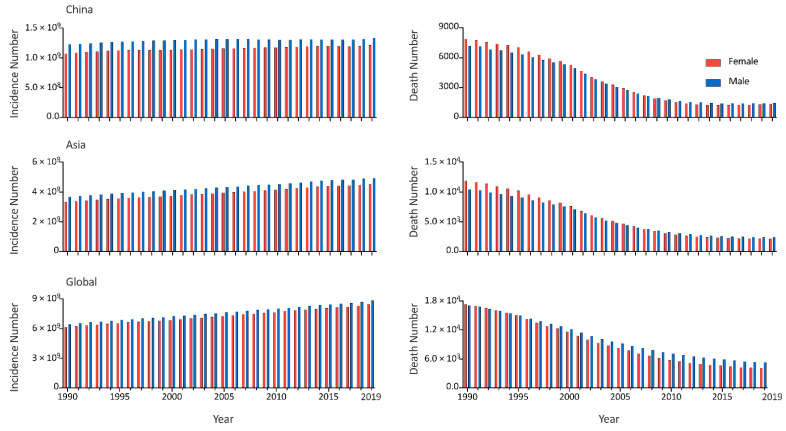
The number of upper respiratory infection cases and deaths by sex, from 1990 to 2019, in China, Asia, and the rest of the globe.

**Figure 4 viruses-14-02550-f004:**
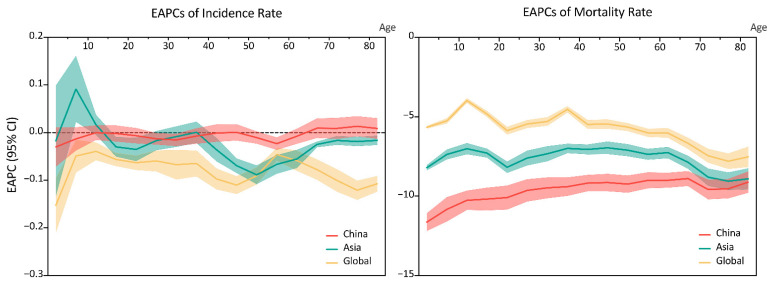
EAPCs of age-specified incidence and mortality rates of upper respiratory infections from 1990 to 2019, in China, Asia, and the rest of the globe. EAPC, estimated annual percentage change.

**Figure 5 viruses-14-02550-f005:**
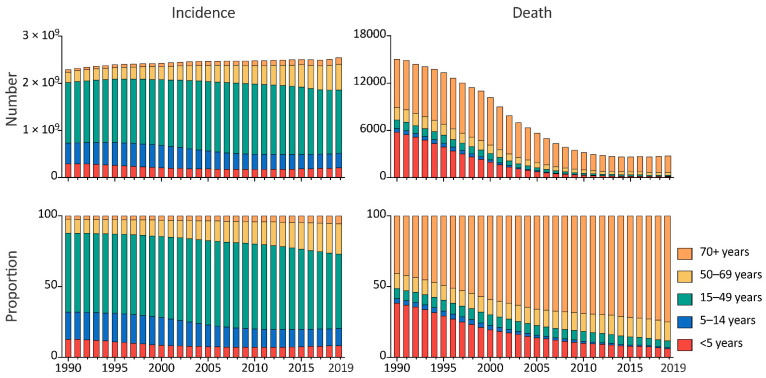
The number and proportions of upper respiratory infection cases and deaths by age group, from 1990 to 2019, in China.

**Figure 6 viruses-14-02550-f006:**
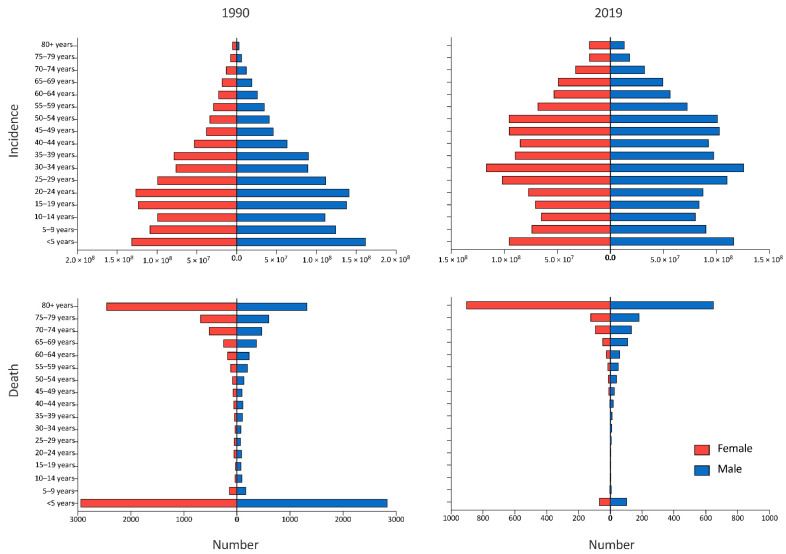
The age and sex distribution of upper respiratory infection cases and deaths in 1990 and 2019, in China.

**Figure 7 viruses-14-02550-f007:**
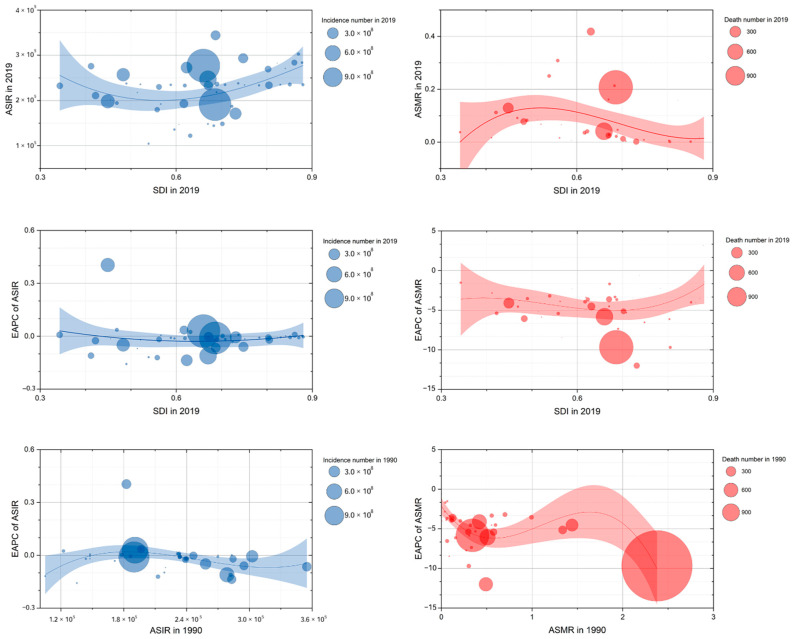
The correlation between ASIR/ASMR of upper respiratory infections and SDI (in 2019), EAPC of ASIR/ASMR and SDI (in 2019), and EAPC of ASIR/ASMR and ASIR/ASMR (in 1990), in Asian countries. ASIR, age-standardized incidence rate; ASMR, age-standardized mortality rate; SDI, Socio-demographic Index; EAPC, estimated annual percentage change.

## Data Availability

Not applicable.

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
