# Peer review of "Incidence and Mortality Trends of Upper Respiratory Infections in China and Other Asian Countries from 1990 to 2019"

_viruses, 2022, doi:10.3390/v14112550_

Round 1
Reviewer 1 Report
In this review, author showed the current status and burden of upper respiratory infections (URIs) in Asian countries from 1990 to 2019, using the data from the Global Burden of Diseases Study 2019 results. Absolute incidence and death number represented the actual situation of URI in each country, so author estimated age-standardized incidence, and mortality rate (ASIR and ASMR), and age and sex specified incidence and mortality rates of URIs in Asian countries. This review concludes, Thailand had the highest ASIR of URI both in 1990 and in 2019; and the highest ASMR was in China in 1990, and in Uzbekistan in 2019. Children under the age of 5 had the highest incidence rate and the second highest mortality rate, while the highest mortality rate of URI was in the elderly. It is clearly a big effort of author for doing this large data study, but despite of positive outcomes I have some suggestion where it can be improved.
Comments
1. Although author did a great effort in this review, but few papers are already published showing national burden and trends of upper respiratory infections (URIs) in Asian countries. In 2021, a review published (X. Jin et al. / EClinicalMedicine 37 (2021) 100986) showing the Global burden of upper respiratory infections in 204 countries and territories, from 1990 to 2019. This review also points out all the factors related to URI globally. Author did not add this paper as a reference also. Can you please add this reference.
2. Author mainly focused on China and compare all factors, like age and sex only in China, whereas other countries in Asia also have ASIR and ASMR, so either author have to discuss more about other countries or changed the title “Incidence and mortality trends of upper respiratory infections in China from 1990 to 2019”
3. Can you please add the medical history of the population like they are smoker or not as well the major cause of this infection. I know you mentioned influenza virus may result URI, but you can add all the critical points delated to this infection.
4. Figure 7 is not clear, it looks blurry.
5. Overall, the study is preliminary, and author need to perform more analysis, so that it can be useful for the readers.
6. In discussion, I did not find any novelty. I think author should need to work on their discussion section.

Author Response
Reviewer 1:
In this review, author showed the current status and burden of upper respiratory infections (URIs) in Asian countries from 1990 to 2019, using the data from the Global Burden of Diseases Study 2019 results. Absolute incidence and death number represented the actual situation of URI in each country, so author estimated age-standardized incidence, and mortality rate (ASIR and ASMR), and age and sex specified incidence and mortality rates of URIs in Asian countries. This review concludes, Thailand had the highest ASIR of URI both in 1990 and in 2019; and the highest ASMR was in China in 1990, and in Uzbekistan in 2019. Children under the age of 5 had the highest incidence rate and the second highest mortality rate, while the highest mortality rate of URI was in the elderly. It is clearly a big effort of author for doing this large data study, but despite of positive outcomes I have some suggestion where it can be improved.
Comments
- Although author did a great effort in this review, but few papers are already published showing national burden and trends of upper respiratory infections (URIs) in Asian countries. In 2021, a review published (X. Jin et al. / EClinicalMedicine 37 (2021) 100986) showing the Global burden of upper respiratory infections in 204 countries and territories, from 1990 to 2019. This review also points out all the factors related to URI globally. Author did not add this paper as a reference also. Can you please add this reference.
Response: Thanks for the reviewer’s suggestion. We have added this reference as shown in Line 131 to 132: A recent study had assessed the global burden of URIs, mostly discussing regional URIs, but had not emphasized the great URI burden in Asian countries.
- Author mainly focused on China and compare all factors, like age and sex only in China, whereas other countries in Asia also have ASIR and ASMR, so either author have to discuss more about other countries or changed the title “Incidence and mortality trends of upper respiratory infections in China from 1990 to 2019”
Response: Thanks for the reviewer’s suggestion. We have changed the title into “Incidence and mortality trends of upper respiratory infections in China and other Asian countries from 1990 to 2019”.
- Can you please add the medical history of the population like they are smoker or not as well the major cause of this infection. I know you mentioned influenza virus may result URI, but you can add all the critical points delated to this infection.
Response: Thanks for the reviewer’s suggestion. We have revised the sentences as shown in line 33 to 39: Respiratory infections remain a major public health around the world. Infections of the upper respiratory tract, such as laryngitis, pharyngitis, nasopharyngitis, and rhinitis, are among the most common diseases in primary medical care. Respiratory infections caused by influenza affects people in all age groups, resulting 3 to 5 million cases of severe illness, and about 290,000 to 650,000 deaths annually across the world. Lower respiratory infections (LRI) caused by environmental tobacco smoke exposure were also an important threat to people’s health, especially for children and infants.
- Figure 7 is not clear, it looks blurry.
Response: Thanks for the reviewer’s suggestion. We have changed it into clearer one.
- Overall, the study is preliminary, and author need to perform more analysis, so that it can be useful for the readers.
Response: Thanks for the reviewer’s suggestion. The editor suggested us to change our paper from article into communication, and we had to limit the length of our manuscript. Nevertheless, we descripted the current burden and long-term changes of URIs in Asian countries, assessed the estimated annual percentage changes (EAPC) of the incidence and mortality rates of URIs throughout the past three decades, and also explore the factors that associated with EAPC and ASIR/ASMR.
- In discussion, I did not find any novelty. I think author should need to work on their discussion section.
Response: Thanks for the reviewer’s suggestion. We have revised the discussion section. Our findings were consistent with previous studies, which enhanced the cognition of URI burden and age/sex distribution in Asian countries. Moreover, the factors that associated with EAPC and ASIR/ASMR in our study added the understanding of URIs, which could help to make national or regional strategies for prevention and control of respiratory infections.
Reviewer 2 Report
This paper deals with a significant clinical and public health issue, the burden of upper respiratory infection in Asian countries from 1990 to 2019. The methods seem appropriate. The writing needs some improvement. The results are significant.
An important issue that needs to be rectified relates to the true aim of this study and the relation between upper respiratory infections (URI) in general and influenza in specific. The title does not mention influenza. But, the first sentence of the abstract is about respiratory infection caused by influenza virus. The aim is " to assess the current status and trends of URI burden from 1990 to 2019 in Asian countries". It shouldn't be too hard to rectify this. If the references to influenza early in the paper are to remain, greater effort should be made in the background and discussion, to consider the proportion of URIs attributable to influenza virus, and variations in this proportion through time and between countries. Additional clarification should be given of the impacts of vaccination for influenza on URI rates should be provided, as this is only mentioned in passing.
It's appropriate that air pollution is mentioned. But smoking and vaping, the former particularly, certainly have an important association with respiratory tract infections. Smoking prevalence by country, at least, rates mention and discussion. Some of the countries studied have high smoking prevalence.
Figure 1 contains such a large amount of information that it's not readily comprehensible. I'd recommend including a limited selection of the data from it, with the complete figure in the supplementary materials.
The paper should be edited by a native English speaker. While meaning is generally apparent there is some unusual usage (e.g. "However, there lacks studies that...", line 54). Minor typographical errors need to be fixed (e.g. "rated" should be "rates", line 170; "remains" should be "remain" in the first sentence of the abstract).
Author Response
Reviewer 2:
This paper deals with a significant clinical and public health issue, the burden of upper respiratory infection in Asian countries from 1990 to 2019. The methods seem appropriate. The writing needs some improvement. The results are significant.
- An important issue that needs to be rectified relates to the true aim of this study and the relation between upper respiratory infections (URI) in general and influenza in specific. The title does not mention influenza. But, the first sentence of the abstract is about respiratory infection caused by influenza virus. The aim is " to assess the current status and trends of URI burden from 1990 to 2019 in Asian countries". It shouldn't be too hard to rectify this. If the references to influenza early in the paper are to remain, greater effort should be made in the background and discussion, to consider the proportion of URIs attributable to influenza virus, and variations in this proportion through time and between countries. Additional clarification should be given of the impacts of vaccination for influenza on URI rates should be provided, as this is only mentioned in passing.
Response: Thanks for the reviewer’s suggestion. We have revised and made the background section more suitable for the aim of our study. See the introduction: “Respiratory infections remain a major public health around the world. Infections of the upper respiratory tract, such as laryngitis, pharyngitis, nasopharyngitis, and rhinitis, are among the most common diseases in primary medical care. Respiratory infections caused by influenza affects people in all age groups, resulting 3 to 5 million cases of severe illness, and about 290,000 to 650,000 deaths annually across the world. Lower respiratory infections (LRI) caused by environmental tobacco smoke exposure were also an important threat to people’s health, especially for children and infants. As home to nearly 60% of the global population, and including three of the world’s most populous countries in China, India and Indonesia, Asia could be well positioned to exert a significant influence on global prevention and control of respiratory infections. However, there is a lack of studies that analyze respiratory infection burden in all Asian countries, due to the lack of high-quality surveillance data of influenza and other virus infections. …”
- It's appropriate that air pollution is mentioned. But smoking and vaping, the former particularly, certainly have an important association with respiratory tract infections. Smoking prevalence by country, at least, rates mention and discussion. Some of the countries studied have high smoking prevalence.
Response: Thanks for the reviewer’s suggestion. We have added the discussion as shown in line 375 to 410: Besides air pollution, smoking and vaping also have an important association with respiratory tract infections. Previous study showed that most countries, which had significantly higher smoking prevalence than the global average for man, were located in southeast Asia; and China, India, and Indonesia had the largest three smoking populations. Environmental tobacco smoke exposure could cause LRIs and affects people’s livelihoods, especially for children and infants. Studies showed that E-cigarette exposure could disrupt pulmonary homeostasis, by disturbing gas exchange, reducing lung function, increasing airway inflammation and oxidative stress, downregulating immunity, and increasing risk of respiratory infection.
- Figure 1 contains such a large amount of information that it's not readily comprehensible. I'd recommend including a limited selection of the data from it, with the complete figure in the supplementary materials.
Response: Thanks for the reviewer’s suggestion. Figure 1 contains heatmaps that are easily for readers to find out which countries were at the most worrisome situation among all the Asian countries, such as the highest ASIR and ASMR, the largest incidence and death number (except the global and Asian level), and the speediest increasing of URIs. Therefore, we kept Figure 1 in our paper.
- The paper should be edited by a native English speaker. While meaning is generally apparent there is some unusual usage (e.g. "However, there lacks studies that...", line 54). Minor typographical errors need to be fixed (e.g. "rated" should be "rates", line 170; "remains" should be "remain" in the first sentence of the abstract).
Response: Thanks for the reviewer’s suggestion. We have re-read the paper and revised the unusual usage and typographical errors.
Round 2
Reviewer 1 Report
Thank you for doing these changes. It is clearly a big effort of author for doing this large data study.